# Detection of apoptosis and matrical degeneration within the intervertebral discs of rats due to passive cigarette smoking

**Masahiro Nakahashi[1,2], Mariko Esumi[2,3]\*, Yasuaki Tokuhashi[1,2]**

**1** Department of Orthopaedic Surgery, Nihon University School of Medicine, Itabashi-ku, Tokyo, Japan,
**2** Department of Therapeutics for Aging Locomotive Disorders, Nihon University School of Medicine, Itabashi-ku, Tokyo, Japan, **3** Department of Biomedical Sciences, Nihon University School of Medicine, Itabashi-ku, Tokyo, Japan

\* esumi.mariko@nihon-u.ac.jp

**Data Availability Statement:** All relevant data are within the manuscript and its Supporting Information files.

## Abstract

Although low-back pain is considered to be associated with cigarette smoking, the influence of cigarette smoking on the intervertebral discs (IVD) has not been confirmed. We established a rat model of passive cigarette smoking-induced IVD degeneration, and investigated the cytohistological changes in the IVD and the accompanying changes in gene expression. IVD from rats exposed to 8 weeks of passive cigarette smoking were stained with Elastica van Gieson, and exhibited marked destruction of the supportive structure of the reticular matrix in the nucleus pulposus (NP). Positive signals on safranin O, alcian blue, type II collagen and aggrecan staining were decreased in the destroyed structure. Safranin O and type II collagen signals were also decreased in the cartilage end-plate (CEP) after 4- and 8-weeks of cigarette smoking. In the CEP, the potential for apoptosis was increased significantly, as demonstrated by staining for single-strand DNA. However, there were no signs of apoptosis in the NP or annulus fibrosus cells. Based on these findings, we hypothesized that passive cigarette smoking-induced stress stimuli first affect the CEP through blood flow due to the histological proximity, thereby stimulating chondrocyte apoptosis and reduction of the extracellular matrix (ECM). This leads to reduction of the ECM in the NP, destroying the NP matrix, which can then progress to IVD degeneration.

## Introduction

Low-back pain is a highly prevalent disease, and a major problem for performing activities of daily living and health economics. Many studies have reported that cigarette smoking is a risk factor for low-back pain; i.e., 70% of persons with low-back pain are cigarette smokers, and the frequency, amount and duration of cigarette smoking correlate with the incidence of low-back pain [1–9]. In contrast, smoking cessation has been reported to improve patient-reported pain and is related to increased fusion rates [10, 11]. Moreover, cigarette smoking is also associated with degenerative disc disease, including middle-aged disc herniation [12–15]. Cigarette smoking leads to the formation of carboxy-hemoglobin [16], vasoconstriction [16] and

**Funding:** The Department of Therapeutics for Aging Locomotive Disorders is funded by a combination of grants and individual contributions. Funding for this study was provided by charitable donations from Ono Pharmaceutical Co., Ltd., Stryker Japan K. K. and Nakashima Medical Co., Ltd. The funders had no role in study design, data collection and analysis, decision to publish, or preparation of the manuscript.

**Competing interests:** This work was funded in part by Ono Pharmaceutical Co., Ltd., Stryker Japan K. K. and Nakashima Medical Co., Ltd. received by YT. There are no patents, products in development or marketed products associated with this research to declare. This does not alter the authors' adherence to PLOS ONE policies on sharing data and materials.

arteriosclerosis [16, 17], and thus decreases oxygen transport and blood flow [16, 17]. These events are considered to lead to malnutrition of the intervertebral discs (IVD) and promote IVD degeneration. Studies using animal models and *in vitro* culturing of disc cells suggested that nicotine and tobacco smoke exposure induces degenerative changes in the spine [18–22]. Although there have been significant advances in our understanding of the biology underlying IVD degeneration [23] [24], the molecular mechanisms underlying the IVD changes induced by cigarette smoking remain to be elucidated.

To directly clarify the influence of passive cigarette smoking on IVD, we established a rat model of passive cigarette smoking. In passive cigarette smoking rats, slight structural changes were noted on haematoxylin-eosin and alcian blue + periodic acid-Schiff staining of the IVD tissue [25], and the expression of type I and IX collagen mRNA was reduced [26]. Comprehensive investigation with gene expression microarrays revealed increased expression of heat shock protein 70 and protein tyrosine phosphatase [27]. The expression of these genes was also increased in the isolated nucleus pulposus (NP) and annulus fibrosus (AF) [28], suggesting that the passive cigarette smoking-induced stress response occurs similarly in the NP and AF, and induces anti-apoptotic responses. Recently, we observed that passive cigarette smoking also changes the circadian rhythm of clock genes in rat IVD [29]. To further investigate the relationship between the morphological and molecular changes in passive cigarette smoking-induced IVD degeneration, we examined the histological changes and molecular events in the extracellular matrix (ECM) and chondrocytes of the IVD. Based on these findings, we proposed a model of passive cigarette smoking-induced IVD degeneration in rats.

## Materials and methods

### Animals

Twelve male, 8-week-old Sprague-Dawley rats (CLEA Japan, Inc., Tokyo, Japan) were subjected to passive cigarette smoking using our cigarette smoking device, which has been described previously [26, 29]. During housing, the rat health status was monitored twice daily. The monitoring items were food and water intake difficulty, anxiety symptoms such as self-mutilation, respiratory disorder and vocalization, and weight loss without rapid recovery. No rats showed these symptoms during the experiment. Rats underwent passive cigarette smoke exposure for 4 or 8 weeks, designated as the S4 and S8 groups, respectively (n = 6/group). As the respective control groups, non-smoking control rats were established as the N4 and N8 groups, respectively (n = 6/group).

Euthanasia was performed under deep anaesthesia with intraperitoneal administration of 30 mg of pentobarbital sodium and efforts were made to minimize the animal's discomfort. After euthanasia, a longitudinal incision was made immediately above the dorsal spine and the entire spine was excised. The lumbar IVD excluding CEP were separated from the vertebrae and immediately frozen at -80˚C. These samples were used for mRNA measurement and Western blotting. The lower thoracic vertebrae were also excised *en bloc*, fixed in 10% formalin, decalcified with EDTA for 2 months and embedded in paraffin. The study protocol was approved by the Animal Experimentation Committee of the Nihon University School of Medicine.

### Quantification of mRNA

Total RNA extraction, cDNA synthesis, and quantitative PCR were performed as described previously [30]. Briefly, total RNA was extracted from two IVDs of each rat. PCR was performed in 20 μl of reaction mixture containing TaqMan Universal PCR master mix (Applied Biosystems, Foster, USA), TaqMan Gene Expression Assay probe and primers (S1 Table) and

10 ng of cDNA template by Rotor-Gene 3000 (Corbett Life Science, Mortlake, Australia) at 95˚C for 10 min, followed by 45 cycles of 95˚C for 15 s, 60˚C for 1 min.

## Immunohistochemistry and specific staining

Thin paraffin-embedded sections (4 μm) were cut and treated with 600 U/ml of hyaluronidase (Sigma-Aldrich, St. Louis, USA) at 37˚C for 1 hr to activate antigens. Sections were stained using the CSA II kit (Biotin-free catalysed signal amplification system; DAKO, Glostrup, Denmark) and 3,3'-diaminobenzidine (DAB). Type II collagen was detected using a mouse monoclonal antibody (10 μg/ml; Daiichi Fine Chemical, Toyama, Japan) and aggrecan was detected using a mouse monoclonal antibody (10 μg/ml; Thermo Fisher Scientific, Massachusetts, USA) at 37˚C for 1 hr, followed by incubation with horse radish peroxidase (HRP)-labelled goat anti-mouse IgG (DAKO) at room temperature for 15 min. Haematoxylin was used for counterstaining. To quantify staining, the stained tissues were imaged under a microscope (OLYMPUS BX51, Tokyo, Japan) and the positive areas were measured using Win ROOF version 5.6 (Mitani, Co., Fukui, Japan). To evaluate the staining in each region, the tissue images were divided into the NP, the AF, and the peripheral and central regions of the cartilage endplate (CEP). The percent areas that stained positive were calculated for each region (S1 Fig). Four IVD were observed for each of the 6 animals per group.

The above-described thin sections were subjected to Elastica van Gieson (EVG), safranin O and alcian blue staining, and positivity was evaluated. On safranin O staining, the red-stained area, representing acidic proteoglycan (PG) in the CEP, was measured using Win ROOF Version 5.6, and the percent positive area was calculated for the peripheral and central regions of the CEP (S1 Fig).

## DNA fragmentation

DNA fragmentation was detected by immunohistochemistry using an antibody against single-strand DNA (ssDNA). After blocking with 5% skim milk at 37˚C for 1 hr, the sections were incubated with anti-ssDNA rabbit IgG (Immuno-Biological Laboratories Co., Gunma, Japan) at 37˚C for 1 hr, followed by reaction with HRP-labelled anti-rabbit IgG antibody (Immuno-Biological Laboratories Co.) at 25˚C for 30 min and colour development using DAB. Haematoxylin was used for counterstaining. Four IVD were observed for each of the 6 animals per group. The numbers of ssDNA-positive and -negative cells in the CEP were measured, and the positive rate was calculated by dividing the number of positive cells by the total number of cells. The specificity of the staining was confirmed using the thymus tissues from rats treated with and without dexamethasone.

## Statistical analysis

The Mann-Whitney U test was used to determine the significance of differences between two groups (group N4 vs. group S4, group N8 vs. group S8). Differences were considered significant when the $p$-values were less than 0.05.

# Results

## Histological changes of the IVD induced by passive cigarette smoking

EVG staining was employed to investigate changes in the ECM of the IVD, and specifically demonstrated fibrous structural changes in the NP that were induced by passive cigarette smoking (Fig 1A). As expected, the AF and CEP were covered with collagen fibres, which stained red, and were not affected by passive cigarette smoking. In contrast, no red-staining of

A

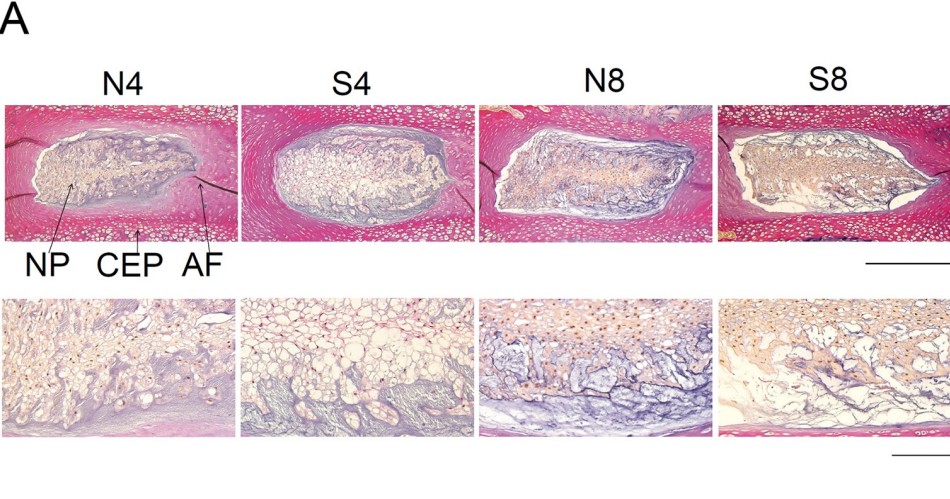

B

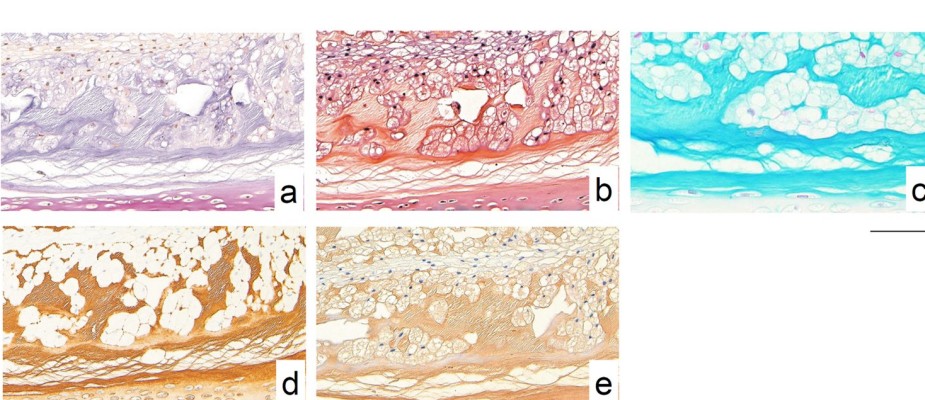

**Fig 1. Histological changes of the rat IVD induced by passive cigarette smoking.** (A) EVG staining of IVD from control non-smoking (N4) and smoking (S4) rats for 4 weeks or 8 weeks (N8 and S8, respectively). Upper and lower panels represent low and high magnification, respectively. Bars in the left and right panels indicate 500 μm and 100 μm, respectively. NP, nucleus pulposus; AF, annulus fibrosus; CEP, cartilage end-plate. (B) Staining of NP of IVD from control non-smoking rat N4. a, EVG staining; b, safranin O staining; c, alcian blue staining; d, immunohistochemical staining for Type II collagen; e, immunohistochemical staining for aggrecan. Bar indicates 100 μm.

the collagen fibres was noted in the NP, and dark purple stained elastic fibres were present in the surrounding region, exhibiting a closed reticular structure. In the central region, NP cells (notochordal cells) containing pink-stained cytoplasm were present in clusters, and cytoplasm that appeared to outline vacuoles was evident in some cells. Passive cigarette smoking markedly altered these characteristic structures of the NP. In particular, marked destruction of the reticular structure and condensation of NP cells were observed after 8-weeks of passive cigarette smoking (Fig 1A). To identify the components of this reticular structure, safranin O and alcian blue staining, and immunostaining for type II collagen and aggrecan were performed for the IVD from the non-cigarette smoking rats (group N4) (Fig 1B). The reticular structures that stained dark purple on EVG staining were positive for all stains (i.e., safranin O, alcian blue, type II collagen and aggrecan). Sulphated polysaccharide-containing PG and type II collagen fibres comprised the interstitium-supportive structure in the NP (Fig 1B), which was

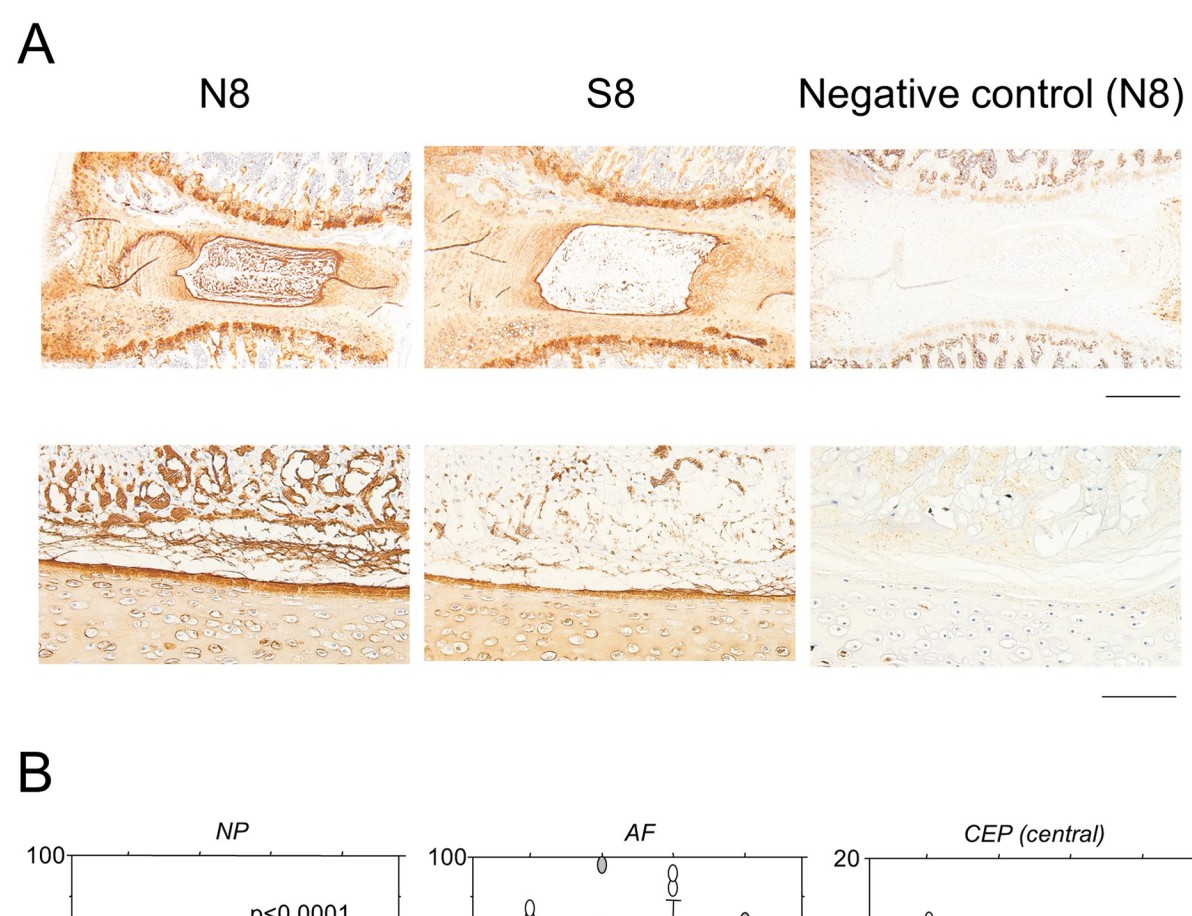

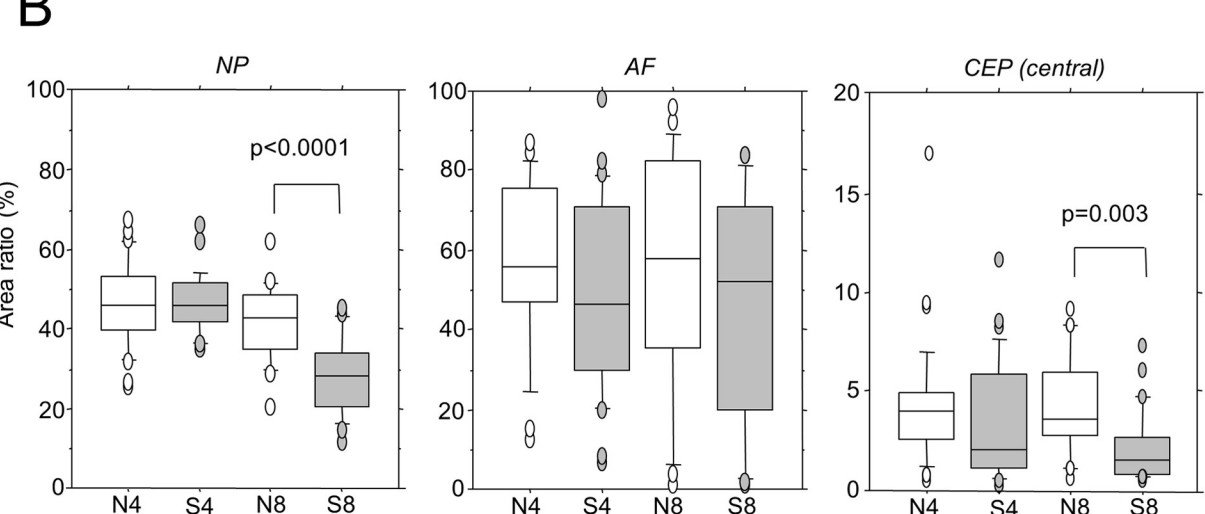

**Fig 2. Immunohistochemical staining for type II collagen.** (A) Immunostaining for Type II collagen in the IVD from control non-smoking (N8) and smoking (S8) rats for 8 weeks. The IVD from N8 were stained without the primary antibody as a negative control. Upper and lower panels are low and high magnification, respectively. Bars indicate 500 μm and 100 μm, respectively. (B) The type II collagen-positive area was measured and the positive rate was calculated. NP, nucleus pulposus; AF, annulus fibrosus; CEP (central), central region of cartilage end-plate. The area of NP, AF and CEP (central) is specified as shown in S1A Fig. *P*-values were determined by the Mann-Whitney U test.

destroyed by passive cigarette smoking (Fig 1A). We therefore investigated how the individual components of the NP interstitium were affected by passive cigarette smoking.

Intense staining for type II collagen was observed in the reticular structure of the NP, which was consistent with the EVG staining (Figs 1B-a, 1B-d and 2A). Type II collagen-positive areas were subsequently evaluated in the NP, and compared between the S8 and N8 groups. Although the percent positive area decreased significantly in the S8 group, no

significant differences were noted in the AF (Fig 2B). Similarly, when the positive areas were measured in the CEP, significant decreases were observed in the central region in the S8 group (Fig 2B). Therefore, type II collagen expression was reduced in the NP and CEP. Synthesis ability was investigated by measuring mRNA expression, but no significant decreases were noted (S2A Fig and S1 Table). Moreover, there were no changes in the expression of the type II collagen-degrading enzyme, Mmp13 (S2A Fig and S1 Table). Intense staining for aggrecan was also observed in the interstitium-supportive structure in the NP, which decreased significantly in the S8 group (Fig 3). Synthesis and degradation of aggrecan were also investigated at the mRNA level. No changes were noted in the expression of aggrecan or Mmp3 mRNA, but Adamts4 mRNA was reduced in both the 4- and 8-week passive cigarette smoking groups (S2B Fig).

Safranin O-stained IVD are shown in Fig 4. Passive cigarette smoking markedly affected the ECM staining of the CEP. The red staining of the acidic PG bound by safranin O was markedly reduced by passive cigarette smoking, whereas the green staining of non-collagen protein bound by fast green became more apparent. The quantitative results are shown in Fig 4B. The percent PG positive area decreased significantly in all regions of the CEP in both the 4- and 8-week passive smoking groups compared with non-smoking controls. This led us to question the mechanism by which the PG decreased in the CEP in the early stage. Focusing on functional inactivation of chondrocytes in the CEP, we subsequently investigated apoptosis in these cells.

## Induction of apoptosis in the CEP by passive cigarette smoking

Apoptosis was investigated using immunohistochemical staining for ssDNA. ssDNA-positive cells were present in the IVD tissues of healthy rats (S3 Fig). Specifically, 60% and 25% of chondrocytes were positive in the CEP in the S8 and N8 groups, respectively, demonstrating a significant increase in the S8 group. Similarly, the number of positive cells increased significantly in the S4 group compared with the N4 group (Fig 5). In contrast, no passive cigarette smoking-induced changes were noted in the NP or AF cells, although positive cells were present (S3 Fig). The lack of increase in apoptotic reactions was also confirmed by Western blotting for β-actin in the NP and AF (S4 Fig). Fragmentation of β-actin by caspase 3 was evident even in healthy IVD tissue (NP and AF); however, no significant increases in response to passive cigarette smoking were found by quantitation of the fragmentation (S4 Fig). Furthermore, no significant changes in apoptosis-related gene expression were observed in the NP or AF in response to passive cigarette smoking (S1 Table). Apoptosis was observed to some extent even in healthy IVD cells, which was due to the disappearance and replacement of notochordal cells by chondrocyte-like cells with aging [31].

## Discussion

Using this model, we previously observed that rat body weight gain was suppressed by passive cigarette smoking (S5 Fig). A similar effect was observed in rats allowed to self-administer nicotine, independent of food intake, which corresponded to 15 to 60 micrograms/kg/infusion nicotine [32]. We also measured blood nicotine levels in our previous study: 36.5 to 124.8 ng/ml (mean: 72.1 ng/ml) [25]. This concentration range corresponds to 10 to 70 ng/ml (mean: 33 ng/ml) reported for 330 human smokers who smoked 20.7 cigarettes/day on average [25, 33]. Therefore, this model is comparable to humans who are smoking 20 to 40 cigarettes/day. Under these exposure conditions, the NP architecture was destroyed by passive cigarette smoking and the supporting structure composed of the ECM had degenerated based on EVG staining. This supportive structure comprised type II collagen and PG, which were both

A

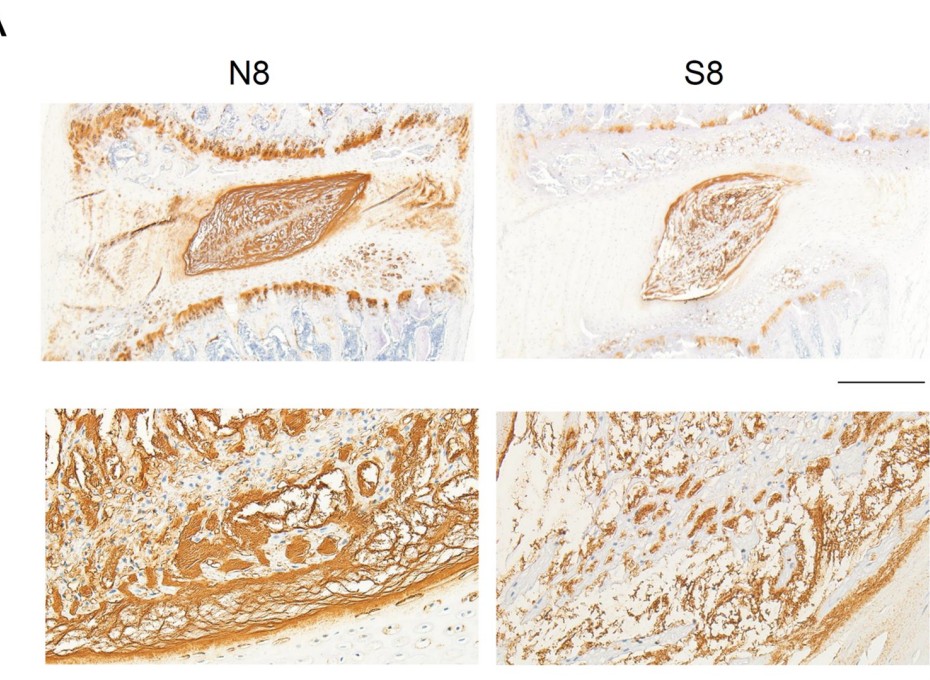

B

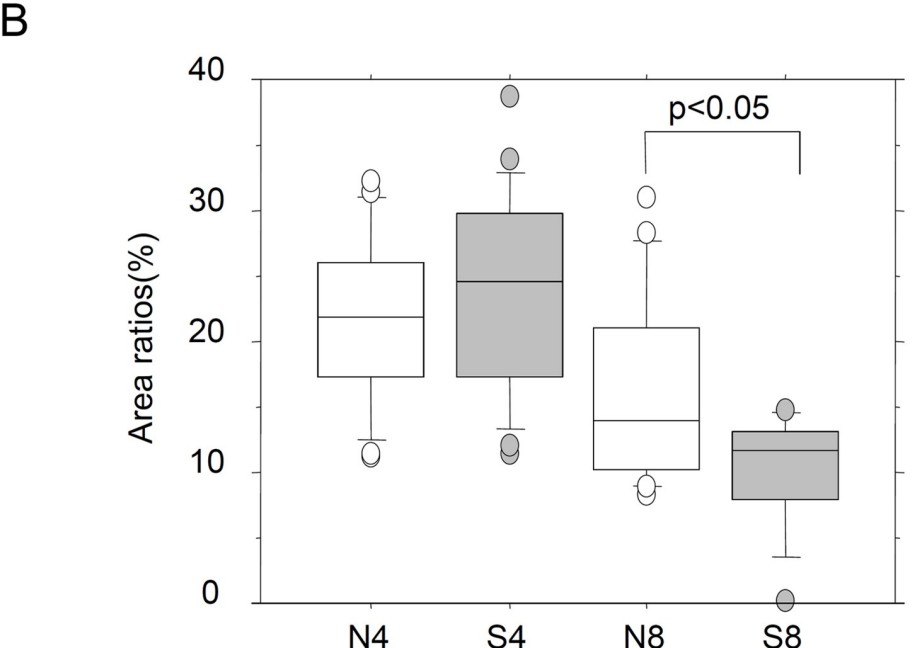

**Fig 3. Immunohistochemical staining for aggrecan.** (A) Immunostaining for aggrecan in the IVD from control non-smoking (N8) and smoking (S8) rats for 8 weeks. Upper and lower panels represent low and high magnification, respectively. Bars indicate 500 μm and 100 μm, respectively. (B) The aggrecan-positive area of the NP (nucleus pulposus) was measured and the positive rate was calculated. The area of NP is specified as shown in S1B Fig. *P*-values were determined by the Mann-Whitney U test.

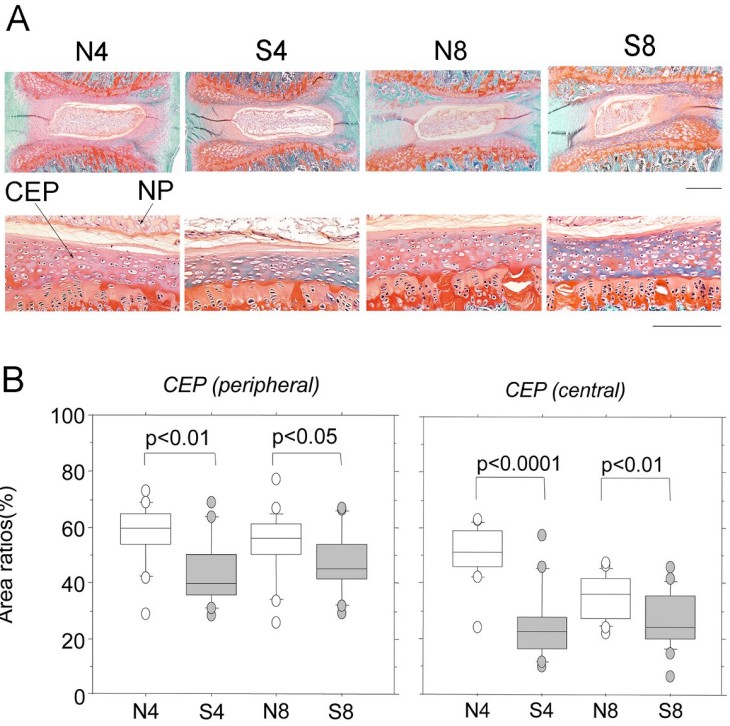

**Fig 4. Safranin O staining of rat IVD and CEP.** (A) Representative histological features from control non-smoking (N4) and smoking (S4) rats for 4 weeks or 8 weeks (N8 and S8, respectively). Upper and lower panels represent low and high magnification, respectively. Bars indicate 500 μm and 100 μm, respectively. NP, nucleus pulposus; CEP, cartilage end-plate. (B) The safranin O-positive area of the CEP was measured and the positive rate was calculated. The CEP was divided into peripheral and central regions, and the safranin O-positive area was measured. CEP (peripheral, central), peripheral or central region of the CEP specified as shown in S1C Fig. *P*-values were determined by the Mann-Whitney U test.

destroyed by passive cigarette smoking. Type II collagen and aggrecan were decreased at the protein level, but this was not supported by quantitative mRNA analysis. Although there were no increases of *Mmp13* or *Mmp3* mRNA in this study, *Adamts4* expression was slightly attenuated in the cigarette smoking groups. Activation of other degrading enzymes may have been involved in the decrease of these matrix proteins. Recently, Ngo *et al.* demonstrated that ADAMTS5 is the primary aggrecanase mediating smoking-induced IVD degeneration in mouse models of chronic tobacco smoking using *ADAMTS5*-deficient mice [34]. Thus, ADAMTS5 may also be the enzyme responsible for the degradation in the current study. Wang *et al.* reported marked loss of disc matrix in a mouse cigarette smoking model using direct smoke inhalation [21]. As their model utilized direct vs. passive inhalation, the smoke conditions were more severe than those used in our study, and the degradation of aggrecan, and reduced synthesis of PG and collagen were also demonstrated. Thus, these findings support the conclusion that tobacco smoke alone is sufficient to affect peripheral tissues and lead to IVD degeneration.

PG and type II collagen were also decreased in the CEP following passive cigarette smoking. These structural constituents are produced and maintained by chondrocytes, suggesting that passive cigarette smoking inhibited cellular function in the CEP. We demonstrated that apoptotic responses of CEP cells were stimulated by passive cigarette smoking. In the CEP, the stimulation of apoptosis and reduction of the ECM may have both occurred in the early stage after 4 weeks of passive cigarette smoking. In contrast, in the NP, the reduction of the ECM

A

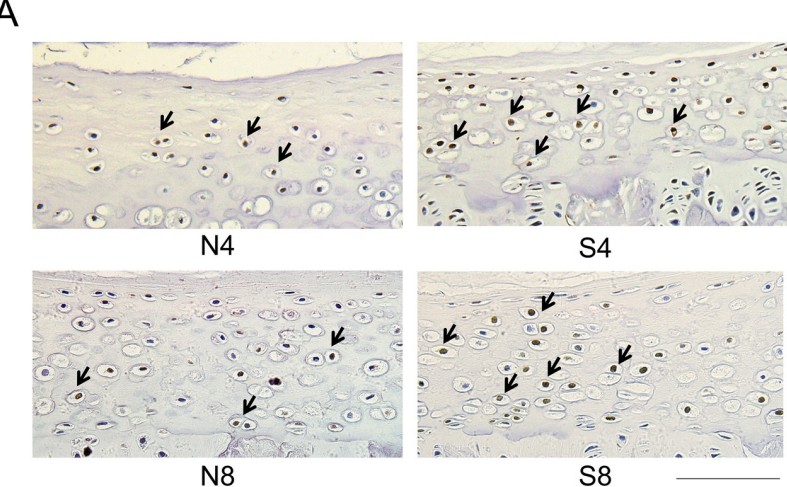

B

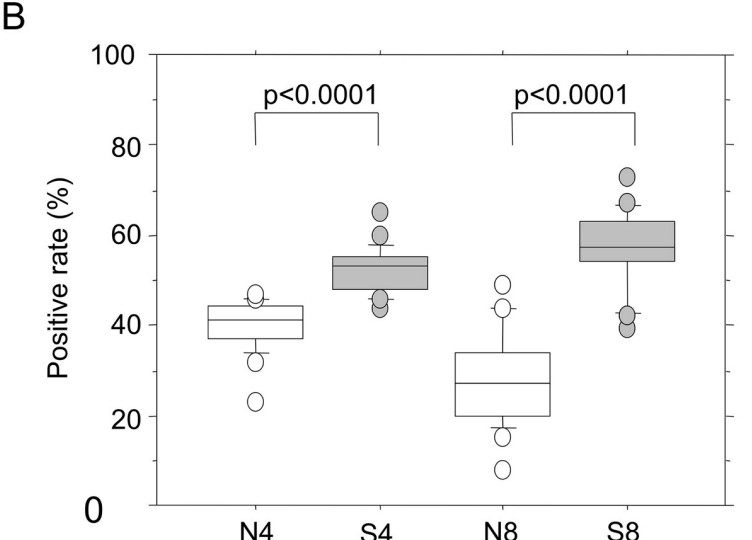

**Fig 5. Apoptotic reaction of the CEP induced by passive cigarette smoking.** (A) Immunostaining for ssDNA of CEP from control non-smoking (N4) and smoking (S4) rats for 4 weeks or 8 weeks (N8 and S8, respectively). Arrows indicate representative cells with ssDNA-positive brown nuclei. The bar indicates 100 μm. (B) The numbers of ssDNA-positive and–negative cells in the CEP were measured and the positive rate was calculated. *P*-values were determined by the Mann-Whitney U test.

was notable by the eighth week. This time lag suggests that early functional changes in the CEP are involved in the changes in the NP ECM. Arana et al. observed that when NP cells were co-cultured with cartilage tissue, expression of PG and type I and II collagen increased in the NP cells, and expression of ECM-degrading enzymes was decreased, suggesting that chondrocytes in the CEP maintain homeostasis of the IVD tissue [35]. Similarly, in our study, the passive cigarette smoking-induced dysfunction of CEP cells may have led to the decrease in type II collagen and PG, and caused changes in the NP or changes in the architecture through NP cells. Ariga *et al*. also reported similar findings, i.e., aging-induced apoptosis and destruction of the CEP structure in mouse IVD, followed by NP and IVD degeneration [36]. Wang *et al*. reported that Fas receptor expression and apoptotic cells were increased in the CEP in addition

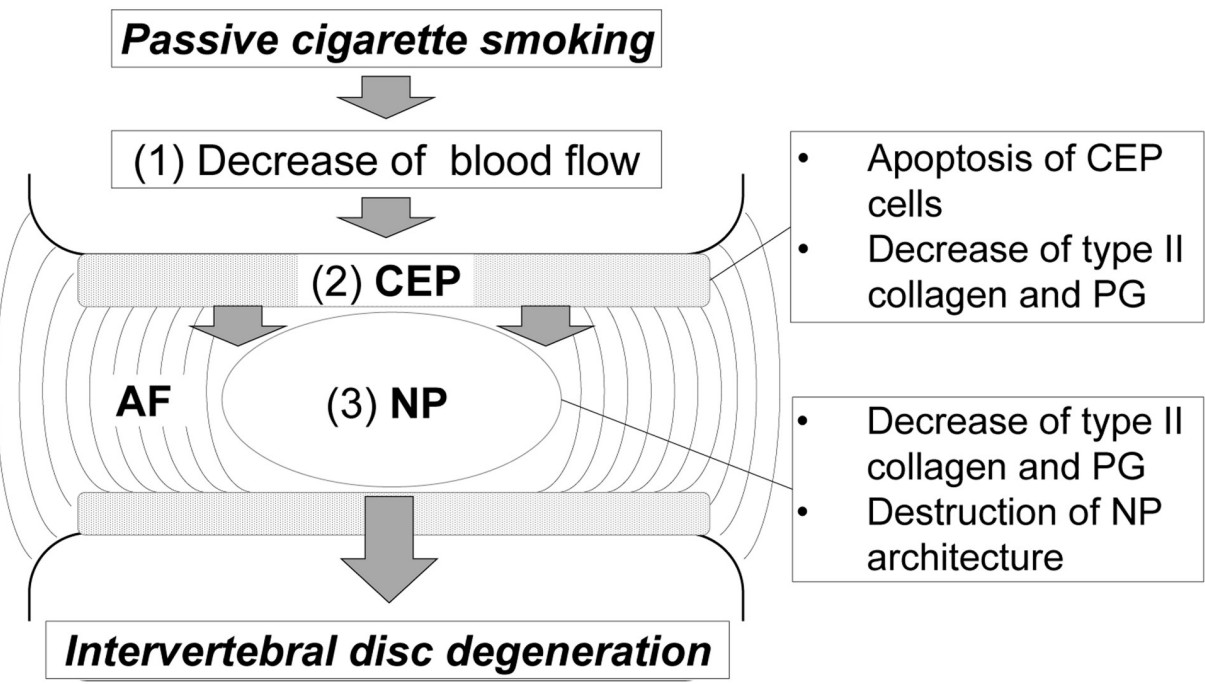

**Fig 6. Schema of the molecular mechanisms underlying IVD degeneration induced by passive cigarette smoking in rats.**

to degeneration of human IVD [37]. These studies support the involvement of apoptosis of chondrocytes in the CEP during the course of IVD degeneration.

Based on this study, we hypothesized the following molecular mechanisms underlying IVD degeneration (Fig 6): 1) Passive cigarette smoking reduces blood flow, which most significantly influences the CEP. 2) Consequently, the potential for apoptosis is increased, and type II collagen and PG levels decrease around the chondrocytes. 3) This influence is transmitted to the NP, leading to further reduction of the production of type II collagen and PG. At the same time, structural changes of the NP cells and destruction of the tissue structure occurs. Regarding 1), a porcine model of cigarette smoking shows a significant reduction in solute transport from the blood capillaries in the IVD [38]. In our previous study on a rabbit model, nicotine treatment results in delineation of vascular buds in the vicinity of the vertebral endplate and a reduction of their numbers [18]. it has been reported that cigarette smoking induces carbon monoxide production, which promotes degradation of hypoxia inducible factor-1 (HIF-1α) and inhibits vascularization [39, 40]. Blood flow into the IVD may be likely decreased in our passive cigarette smoking rat model due to vasoconstriction induced by nicotine. Regarding 2), it is well known that hypoxia and ischemia can cause apoptosis [41]. HIF-2α regulates Fas-mediated chondrocyte apoptosis during osteoarthritic cartilage destruction [42], and the expression of HIF-1α has been reported to correlate significantly with apoptosis in human herniated discs [43]. As such, reduced blood flow-induced hypoxia may also have induced chondrocyte apoptosis in the CEP in our model. However, we did not assess mRNA expression in isolated CEP cells. Thus, further analyses of apoptosis-related gene expression in isolated CEP cells may more clearly demonstrate such changes, as observed in IVD cells. Regarding 3), as similar findings have been reported [36], apoptosis of the CEP may lead to degeneration of the NP and IVD. Wang *et al.* suggested that the occurrence of Fas-mediated apoptosis, which is promoted within the CEP, is not unidirectional, but indeed represents mutual interactions between these tissues and cells [37]. Recently, Elmasry *et al.* suggested that both direct and

indirect effects of smoking play significant roles in IVD degeneration: the nicotine-mediated down-regulation of cell proliferation and anabolism mainly affects GAG levels in the CEP, and the reduction of solute exchange between blood vessels and disc tissue mainly affects GAG levels and cell density in the NP [24]. Thus, there are possible alternative mechanisms responsible for the effects of cigarette smoke on the CEP and NP: the direct effects of nicotine on the CEP and the reduction of transport of nutrients through the CEP to the NP.

We demonstrated the possibility of chondrocyte apoptosis within the CEP of rat IVD in response to passive smoking. Changes were accompanied by decreases in type II collagen and PG in the NP, leading to destruction of the NP architecture. Apoptosis was suggested by the detection of chondrocytes that were positive for ssDNA; however, definitive morphological features of apoptosis were not observed in this study [43]. Therefore, further studies are needed to elucidate the extent of true apoptosis within this region of the IVD in rats exposed to passive cigarette smoking.

## Supporting information

**S1 Fig. Measurement area for histological examination.** (A) Immunohistochemistry for type II collagen. (B) Immunohistochemistry for aggrecan. (C) Safranin O staining. The upper and lower sides of each panel represent the cranial and caudal directions, respectively. The left and right sides of each panel represent the ventral and dorsal directions, respectively. The positive NP area ratio was the ratio of the positive area to the entire NP, and the positive AF area ratio was the ratio of the positive area to the dorsal AF with a diameter of 300 μm. The CEP was divided into the peripheral and central regions, and the positive area percentage was calculated in each region. The bar indicates 200 μm.
(PDF)

**S2 Fig. Quantitative mRNA analysis.** (A) Type II collagen and Mmp13 mRNA. (B) Aggrecan, Mmp3, and Adamts4 mRNA. Two IVD were excised from each rat and homogenized in TRIZOL (Invitrogen, Carlsbad, USA) to extract total RNA. The extracted RNA was treated with DNase I, and then reacted with random hexamer primers and Prime Script reverse transcriptase (TAKARA, Kyoto, Japan) at 30˚C for 10 min, 45˚C for 60 min, and 70˚C for 15 min to synthesize cDNA. Using the TaqMan Gene Expression Assay (Applied Biosystems, Foster, USA), PCR was performed according to the manufacturer's instructions using a Rotor-Gene 6000 real-time analyzer (Corbett Life Science QIAGEN, Alabama, USA). 18S rRNA was measured as an endogenous control for correction of the gene expression levels. For quantification, the absolute quantification method was employed, in which a calibration curve was prepared for each gene from 5-fold serial dilutions using cDNA with the highest expression level as the standard. The measured expression level of each gene was divided by the measured 18S rRNA expression level to calculate the normalized value. This normalized value was compared between the groups. The significance of the differences was analyzed using the Mann-Whitney U test.
(PDF)

**S3 Fig. Immunostaining for ssDNA in the NP and AF.** Left and right panels represent low and high magnification, respectively. Bars indicate 1 mm and 200 μm, respectively. N4, nonsmoking control for 4 weeks; S4, passive smoking for 4 weeks; N8, non-smoking control for 8 weeks; S8, passive smoking for 8 weeks.
(PDF)

**S4 Fig. Fragmentation of β-actin.** (A) Cleavage sites of caspase 1 and caspase 3 in rat β-actin. The forty-two-kDa β-actin is cleaved by caspases-1 at 2 aspartic acid (Asp) residues at positions

11 and 244, producing a 29-kDa fragment. It is also cut at Asp 244 by caspase 3 to produce a 32-kDa fragment. The thick bar from amino acid residues 1 to 100 indicates the epitope of the anti-β-actin antibody used in this study. (B) Immunoblot analysis of β-actin from the IVD (NP and AF). Three IVD from each of 5 rats were combined, and protein was extracted. The IVD were mechanically ground using a mortar, cooled in liquid nitrogen, and extracted with shaking in 1 ml of guanidine hydrochloride extraction solution (4 M guanidine HCl, 50 mM sodium acetate, 65 mM DTT, 10 mM EDTA, Complete Mini Protease Inhibitor Cocktail (Roche), pH 8.5) at 4°C overnight. After centrifugation at 30,000$(x\,g)$ for 5 min, precipitated collagen fibers were removed, and the supernatant was centrifuged again to remove macromolecular proteins that were 100 kDa or larger using a 100 kDa molecular weight cut off centrifugal filter (Millipore, CA, USA). The filtrate was used as the protein extract. The extract was mixed with 9 volumes of 100% ethanol to precipitate any protein. The precipitate was washed and resolved with 1xSDS-PAGE loading buffer. Forty μg of protein was applied to a 12.5% polyacrylamide gel and electrophoresed, followed by blotting onto a nitrocellulose membrane using the iBlot Dry Blotting System (Carlsbad, USA). The membrane was blocked with 5% skim milk/PBS at room temperature for one hour and then reacted with 0.3 μg/ml of mouse monoclonal anti-β-actin antibody (Abcam, Cambridge, UK) at room temperature for one hour, followed by reaction with 0.02 μg/ml of HRP-conjugated anti-mouse IgG goat antibody at room temperature for 30 minutes. Chemiluminescence was induced using ECL Advance (GE Healthcare, Buckingham, UK) and detected using Light-Capture (Atto, Tokyo, Japan). Two bands corresponding to β-actin, 42- and 32-kDa, were also detected in the IVD in the non-smoking control groups, suggesting that physiological cleavage of β-actin by caspase 3 occurs in the normal rat IVD. N4, non-smoking control for 4 weeks; S4, passive smoking for 4 weeks; N8, non-smoking control for 8 weeks; S8, passive smoking for 8 weeks. (C) Quantification of immunoblotting for β-actin. Immunoblot signals were subjected to molecular weight measurement and quantitative analysis using CS Analyzer 2.0 (Atto) and MultiGauge (FUJI-FILM, Tokyo, Japan), respectively. The proportion of β-actin fragmentation by caspase 3 was calculated. Passive cigarette smoking did not induce any change in the fragmentation rate. (PDF)

**S5 Fig. Rat body weight during passive cigarette smoking (grey) compared with that of non-smoking control (white).** * indicates significant decrease in body weight gain in smoking rats (p<0.05 by Mann-Whitney U test).
(PDF)

**S1 Table. Comparison of mRNA levels in the intervertebral disc (IVD) between passive smoking and non-smoking control rats.**
(DOCX)

## Acknowledgments

We thank Dr. Yoshiaki Kusumi for valuable discussions regarding the histological examination, and Rie Takahashi and Mika Sakamoto for their assistance in tissue staining.

## Author Contributions

**Conceptualization:** Mariko Esumi.

**Data curation:** Masahiro Nakahashi.

**Formal analysis:** Masahiro Nakahashi.

**Funding acquisition:** Yasuaki Tokuhashi.

**Investigation:** Masahiro Nakahashi.

**Methodology:** Masahiro Nakahashi, Mariko Esumi.

**Project administration:** Mariko Esumi.

**Supervision:** Mariko Esumi, Yasuaki Tokuhashi.

**Validation:** Masahiro Nakahashi.

**Writing – original draft:** Masahiro Nakahashi.

**Writing – review & editing:** Mariko Esumi, Yasuaki Tokuhashi.

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
