## [Decision Letter · Decision Letter 0]

13 Jun 2019

PONE-D-19-14928

Detection of apoptosis and matrical degeneration within the intervertebral discs of rats due to passive cigarette smoking.

PLOS ONE

Dear Prof. Esumi,

Thank you for submitting your manuscript to PLOS ONE. After careful consideration, we feel that it has merit but does not fully meet PLOS ONE’s publication criteria as it currently stands. Therefore, we invite you to submit a revised version of the manuscript that addresses the points raised during the review process.

We would appreciate receiving your revised manuscript by Jul 28 2019 11:59PM. To enhance the reproducibility of your results, we recommend that if applicable you deposit your laboratory protocols in protocols.io, where a protocol can be assigned its own identifier (DOI) such that it can be cited independently in the future. For instructions see: http://journals.plos.org/plosone/s/submission-guidelines#loc-laboratory-protocols

We look forward to receiving your revised manuscript.

Kind regards,

Hee-Jeong Im Sampen, PhD

Academic Editor

PLOS ONE

Journal Requirements:

2.  Thank you for stating the following in the Financial Disclosure section:"The Department of Therapeutics for Aging Locomotive Disorders is funded by a combination of grants and individual contributions. Funding for this study was provided by charitable donations from Ono Pharmaceutical Co., Ltd., Stryker Japan K. K. and Nakashima Medical Co., Ltd. The funders had no role in study design, data collection and analysis, decision to publish, or preparation of the manuscript."

We note that you received funding from a commercial source: "Ono Pharmaceutical Co., Ltd" and "Nakashima Medical Co., Ltd"

3.  Please note that all PLOS journals ask authors to adhere to our policies for sharing of data and materials: https://journals.plos.org/plosone/s/data-availability. According to PLOS ONE’s Data Availability policy, we require that the minimal dataset underlying results reported in the submission must be made immediately and freely available at the time of publication. As such, please remove any instances of 'unpublished data' or 'data not shown' in your manuscript and replace these with either the relevant data (in the form of additional figures, tables or descriptive text, as appropriate), a citation to where the data can be found, or remove altogether any statements supported by data not presented in the manuscript.

4. To comply with PLOS ONE submissions requirements, in your Methods section, please provide additional information on the animal research and ensure you have included details on the frequency of welfare monitoring and any efforts to alleviate suffering.

Additional Editor Comments (if provided):

Reviewers' comments:

Reviewer's Responses to Questions

**Comments to the Author**

1. Is the manuscript technically sound, and do the data support the conclusions?

Reviewer #1: Partly

2. Has the statistical analysis been performed appropriately and rigorously? 

Reviewer #1: Yes

3. Have the authors made all data underlying the findings in their manuscript fully available?

Reviewer #1: Yes

4. Is the manuscript presented in an intelligible fashion and written in standard English?

Reviewer #1: Yes

5. Review Comments to the Author

Reviewer #1: In this study, Nakahashi et al. established a rat model of passive cigarette smoking induced IVD degeneration, and investigated the cytohistological changes in the IVD.

They found that passive cigarette smoking could stimulate chondrocyte apoptosis in the end plate and reduce the extracellular matrix (ECM) of in the NP, which can then progress to IVD degeneration. This study was interesting and well written. There are few comments for this study.

1. After euthanasia, the spine samples were harvested and then fixed in formalin or frozen at -80°C. The authors should consider doing the perfusion (first with saline to remove blood cells and followed by formalin to preserve the antigen). After perfusion, the samples would not contain the blood cells, which may affect the immunohistochemical staining. By the way, what are the samples frozen at -80°C use for? Are these samples for mRNA measurement? The authors should clarify.

2. The authors found that type II collagen and aggrecan were decreased in the NP and CEP, while the synthesis of these two proteins were not affected by the smoking, as reflected by the mRNA repression measurement. How did the authors measure the mRNA? The authors should state the specific methods in the Methods section.

3. The evidence about the passive cigarette smoking-induced apoptosis in the CEP was not strong. The authors should add some other experiments, such as TUNEL staining. Also, the authors found that no significant changes in apoptosis-related gene expression were observed in the NP or AF in response to passive cigarette smoking. What about the protein expression? they should consider measuring the related protein expression via immunohistochemical staining.

4. In the conclusion, the authors stated passive cigarette smoking-induced stress stimuli first affect the CEP through blood flow. However, there was no results reflecting the blood flow. How did they prove that the blood flow was affected?

6. PLOS authors have the option to publish the peer review history of their article (what does this mean?). If published, this will include your full peer review and any attached files.

Reviewer #1: No

---

## [Author Response · Author response to Decision Letter 0]

26 Jul 2019

July 26, 2019 

Manuscript ID PONE-D-19-14928

Title: Detection of apoptosis and matrical degeneration within the intervertebral discs of rats due to passive cigarette smoking.

Author: Masahiro Nakahashi et al.

Dear Dr. Joerg Heber, Editor-in-Chief, PLOS ONE

: 

We appreciate the time and efforts by the editor and referees in reviewing this manuscript. We also thank the reviewers for valuable comments and suggestion. We have addressed all issues indicated in the review report, and believed that the revised version can meet the journal publication requirements. Please find our point-by-point responses to the editor’s and reviewer’s comments and the revised manuscript highlighting the revisions made. 

In accordance with the editor’s comment, we attached our revised Competing Interests Statement here: 

This work was funded in part by Ono Pharmaceutical Co., Ltd., Stryker Japan K. K. and Nakashima Medical Co., Ltd. received by YT. There are no patents, products in development or marketed products associated with this research to declare. This does not alter our adherence to PLOS ONE policies on sharing data and materials. 

Sincerely yours,

Mariko Esumi

Department of Biomedical Sciences

Nihon University School of Medicine

Ohyaguchikami-cho, Itabashi-ku

Tokyo 173-8610, Japan

Phone: +81-3-3972-8111 Ext. 2241

Fax: +81-3-3972-8199

E-mail: esumi.mariko@nihon-u.ac.jp

Point-by-point responses to editor’s and reviewer’s comments:

To Editor

Comment 1: 

Response: 

We have reviewed the PLOS ONE style templates and corrected our manuscript.

Comment 2:

We note that you received funding from a commercial source: "Ono Pharmaceutical Co., Ltd" and "Nakashima Medical Co., Ltd"

Within this Competing Interests Statement, please confirm that this does not alter your adherence to all PLOS ONE policies on sharing data and materials by including the following statement: "This does not alter our adherence to PLOS ONE policies on sharing data and materials.”

Response: 

 We have corrected the Competing Interests Statement as follows:

This work was funded in part by Ono Pharmaceutical Co., Ltd., Stryker Japan K. K. and Nakashima Medical Co., Ltd. received by YT. There are no patents, products in development or marketed products associated with this research to declare. This does not alter our adherence to PLOS ONE policies on sharing data and materials. 

Response: 

 We have included our amended COI statement within our cover letter.

Comment 3: 

Please note that all PLOS journals ask authors to adhere to our policies for sharing of data and materials: https://journals.plos.org/plosone/s/data-availability. According to PLOS ONE’s Data Availability policy, we require that the minimal dataset underlying results reported in the submission must be made immediately and freely available at the time of publication. As such, please remove any instances of 'unpublished data' or 'data not shown' in your manuscript and replace these with either the relevant data (in the form of additional figures, tables or descriptive text, as appropriate), a citation to where the data can be found, or remove altogether any statements supported by data not presented in the manuscript.

Response: 

In accordance with the editor’s comment, we have revised the Introduction on page 3, line 67 by adding a reference 27 (written in Japanese) on page 19, lines 475-478. We have also revised Fig 2B by adding a graph of AF; accordingly, we have revised the Result on page 8, line 192-194 and Fig 2B legend on page 8, lines 212 and 213. 

Comment 4: 

To comply with PLOS ONE submissions requirements, in your Methods section, please provide additional information on the animal research and ensure you have included details on the frequency of welfare monitoring and any efforts to alleviate suffering.

Response: 

In accordance with the editor’s comment, we have revised the Materials and Methods on page 4, lines 84-88, and lines 93-94.

 

To Reviewer #1

Comment 1: 

After euthanasia, the spine samples were harvested and then fixed in formalin or frozen at -80°C. The authors should consider doing the perfusion (first with saline to remove blood cells and followed by formalin to preserve the antigen). After perfusion, the samples would not contain the blood cells, which may affect the immunohistochemical staining. 

Response: 

In order to obtain the intact RNA and exactly analyze the mRNA expression, we did not perform the perfusion. Instead, we performed cardiac puncture for blood removal, and quickly excised the entire spine. IVD is　avascular tissue by nature, and there was no problem about the immunohistochemical staining in our experiment. 

By the way, what are the samples frozen at -80°C use for? Are these samples for mRNA measurement? The authors should clarify.

Response: 

Yes, the samples frozen at -80°C were used for mRNA and protein analysis. In accordance with the reviewer’s comment, we have revised the Materials and Methods on page 4, lines 96 and 97. 

Comment 2: 

The authors found that type II collagen and aggrecan were decreased in the NP and CEP, while the synthesis of these two proteins were not affected by the smoking, as reflected by the mRNA repression measurement. How did the authors measure the mRNA? The authors should state the specific methods in the Methods section.

Response: 

In accordance with the reviewer’s comment, we have added the section ‘Quantification of mRNA’ in the Materials and Methods on page 4-5, lines 103-110. We have also added a reference 30 on page 19, lines 487-490.

Comment 3: 

The evidence about the passive cigarette smoking-induced apoptosis in the CEP was not strong. The authors should add some other experiments, such as TUNEL staining. Also, the authors found that no significant changes in apoptosis-related gene expression were observed in the NP or AF in response to passive cigarette smoking. What about the protein expression? they should consider measuring the related protein expression via immunohistochemical staining.

Response: 

We agree with the reviewer that other apoptosis assessments are necessary to confirm the apoptosis of the CEP. Actually, we tried to assess the apoptosis of the CEP by cleaved caspase-3 staining and TUNEL method. Unfortunately, the two methods did not work in our system, even in the positive control specimen, the thymus of rats treated with dexamethasone. It seems to be due to the tissue preparation such as fixation in our experiment. Therefore, we have not done the immunohistochemistry of other apoptosis-related protein. Only the ssDNA staining worked in the positive control tissue and was used here. 

Thus, considering the limitation, we dampened the statement and discussed on the apoptosis of CEP, followed by several supportive data of other groups. 

Comment 4: 

In the conclusion, the authors stated passive cigarette smoking-induced stress stimuli first affect the CEP through blood flow. However, there was no results reflecting the blood flow. How did they prove that the blood flow was affected?

Response:

As the reviewer pointed out, we have no evidence for the reduction of blood flow in our experiment. We have revised the Abstract on page 2, line 36: from ‘concluded’ to ‘hypothesized’.

However, there have been several reports on blood flow reduction in the IVD induced by cigarette smoke and nicotine. We have revised the Discussion on page 13, lines 329-332 and 335 by citing these reports (references 38 and 18). We have also added a reference 38 on pages 22-23, lines 520-523.

---

## [Decision Letter · Decision Letter 1]

5 Aug 2019

Detection of apoptosis and matrical degeneration within the intervertebral discs of rats due to passive cigarette smoking.

PONE-D-19-14928R1

Dear Dr. Esumi,

We are pleased to inform you that your manuscript has been judged scientifically suitable for publication and will be formally accepted for publication once it complies with all outstanding technical requirements.

With kind regards,

Hee-Jeong Im Sampen, PhD

Academic Editor

PLOS ONE

Additional Editor Comments (optional):

Reviewers' comments:

Reviewer's Responses to Questions

**Comments to the Author**

1. If the authors have adequately addressed your comments raised in a previous round of review and you feel that this manuscript is now acceptable for publication, you may indicate that here to bypass the “Comments to the Author” section, enter your conflict of interest statement in the “Confidential to Editor” section, and submit your "Accept" recommendation.

Reviewer #1: All comments have been addressed

2. Is the manuscript technically sound, and do the data support the conclusions?

Reviewer #1: Yes

3. Has the statistical analysis been performed appropriately and rigorously? 

Reviewer #1: N/A

4. Have the authors made all data underlying the findings in their manuscript fully available?

Reviewer #1: Yes

5. Is the manuscript presented in an intelligible fashion and written in standard English?

Reviewer #1: Yes

6. Review Comments to the Author

Reviewer #1: The authors have addressed all the comments that I raised. This paper is interesting. There is no further comments.

7. PLOS authors have the option to publish the peer review history of their article (what does this mean?). If published, this will include your full peer review and any attached files.

Reviewer #1: No

---

## [Editor Report · Acceptance letter]

20 Aug 2019

PONE-D-19-14928R1 

Detection of apoptosis and matrical degeneration within the intervertebral discs of rats due to passive cigarette smoking. 

Dear Dr. Esumi:

I am pleased to inform you that your manuscript has been deemed suitable for publication in PLOS ONE. Congratulations! Your manuscript is now with our production department. 

With kind regards,

on behalf of

Dr. Hee-Jeong Im Sampen 

Academic Editor

PLOS ONE